# Multiomic Profiling Identified EGF Receptor Signaling as a Potential Inhibitor of Type I Interferon Response in Models of Oncolytic Therapy by Vesicular Stomatitis Virus

**DOI:** 10.3390/ijms23095244

**Published:** 2022-05-08

**Authors:** Anastasia S. Nikitina, Anastasia V. Lipatova, Anton O. Goncharov, Anna A. Kliuchnikova, Mikhail A. Pyatnitskiy, Ksenia G. Kuznetsova, Azzam Hamad, Pavel O. Vorobyev, Olga N. Alekseeva, Marah Mahmoud, Yasmin Shakiba, Ksenia S. Anufrieva, Georgy P. Arapidi, Mark V. Ivanov, Irina A. Tarasova, Mikhail V. Gorshkov, Peter M. Chumakov, Sergei A. Moshkovskii

**Affiliations:** 1Federal Research and Clinical Center of Physical-Chemical Medicine, 119435 Moscow, Russia; quokka.smiles@gmail.com (A.S.N.); ulteran@gmail.com (A.O.G.); a.kliuchnikova@gmail.com (A.A.K.); mpyat@mail.ru (M.A.P.); kuznetsova.ks@gmail.com (K.G.K.); anufrieva@phystech.edu (K.S.A.); arapidi@gmail.com (G.P.A.); 2Engelhardt Institute of Molecular Biology, Russian Academy of Sciences, 119991 Moscow, Russia; lipatovaanv@gmail.com (A.V.L.); azzam.hamad@phystech.edu (A.H.); pavel.gealbhain@gmail.com (P.O.V.); alekseeva.on@phystech.edu (O.N.A.); chumakovpm@yahoo.com (P.M.C.); 3Pirogov Russian National Research Medical University, 117997 Moscow, Russia; 4Institute of Biomedical Chemistry, 119121 Moscow, Russia; 5Moscow Institute of Physics and Technology, 141700 Dolgoprudniy, Russia; mrmormah97@gmail.com (M.M.); yasi.shakiba@gmail.com (Y.S.); 6V.L. Talrose Institute for Energy Problems of Chemical Physics, N.N. Semenov Federal Research Center for Chemical Physics, Russian Academy of Sciences, 119334 Moscow, Russia; markmipt@gmail.com (M.V.I.); iatarasova@yandex.ru (I.A.T.); mike.gorshkov@gmail.com (M.V.G.)

**Keywords:** oncolytic virus, vesicular stomatitis virus, glioblastoma, osteosarcoma, type I interferon, epidermal growth factor receptor, human epidermal growth factor receptor 2, gefitinib

## Abstract

Cancer cell lines responded differentially to type I interferon treatment in models of oncolytic therapy using vesicular stomatitis virus (VSV). Two opposite cases were considered in this study, glioblastoma DBTRG-05MG and osteosarcoma HOS cell lines exhibiting resistance and sensitivity to VSV after the treatment, respectively. Type I interferon responses were compared for these cell lines by integrative analysis of the transcriptome, proteome, and RNA editome to identify molecular factors determining differential effects observed. Adenosine-to-inosine RNA editing was equally induced in both cell lines. However, transcriptome analysis showed that the number of differentially expressed genes was much higher in DBTRG-05MG with a specific enrichment in inflammatory proteins. Further, it was found that two genes, EGFR and HER2, were overexpressed in HOS cells compared with DBTRG-05MG, supporting recent reports that EGF receptor signaling attenuates interferon responses via HER2 co-receptor activity. Accordingly, combined treatment of cells with EGF receptor inhibitors such as gefitinib and type I interferon increases the resistance of sensitive cell lines to VSV. Moreover, sensitive cell lines had increased levels of HER2 protein compared with non-sensitive DBTRG-05MG. Presumably, the level of this protein expression in tumor cells might be a predictive biomarker of their resistance to oncolytic viral therapy.

## 1. Introduction

Permanent cell lines derived from malignant tumors are used as models of choice to develop approaches to therapy with oncolytic viruses (OVs). The antiviral response is usually impaired in cancer cells, and therefore OV therapy is considered a promising option for the treatment of neoplasms, alone or in combination with other approaches [1,2,3,4,5]. One example of such a therapy translated to clinics uses a genetically modified Herpes simplex virus I Imlygic, also known as Talimogene laherparepvec. It is used to treat advanced malignant skin melanoma [6]. Among other viruses considered attractive platforms for developing OV strains is vesicular stomatitis virus (VSV), a relatively small single-strand RNA genome member of the rhabdovirus family [7,8].

Although the defects in the interferon signaling observed in cancer cells [9] make it possible to consider viruses as therapeutic agents, tumors differ widely in their antiviral responses. For example, in in vitro experiments, some lines of cancer cells, after treatment with type I interferon, acquire resistance to VSV, while others remain sensitive to the cytolytic effect of the virus [10]. Furthermore, differences in the antiviral activity of interferon were also observed when using other viruses, for example, in hepatoblastoma cell lines treated with the hepatitis C virus [11].

Knowledge of the molecular mechanisms that determine the differences in the sensitivity of tumor cells to viruses is important for developing approaches to cancer therapy using OVs. First, it will help to identify biomarkers of sensitivity to specific OVs, despite the action of type I interferon, to predict the efficiency of viral therapy using biopsy materials of the patients. Second, outside the field of oncology, knowledge about the variants of interferon response disorders can help predict the course of viral diseases in specific patients. Finally, as a theme of the day, Type I interferon signaling is reduced in severe and critical COVID-19 patients [12]. Thus, there is a need in the field to identify molecular biomarkers of normal or impaired antiviral interferon responses in cells beyond cancer therapy.

We tested several cancer cell lines from our collection for sensitivity to VSV after the pretreatment with Type I interferon and selected two cell lines exhibiting polar responses. After treatment with interferon, the glioblastoma cell line DBTRG-05MG [13] acquired resistance to VSV, while the osteosarcoma cell line HOS [14] retained the sensitivity to the virus. In this study, we aimed to find differences in the expression patterns of genes involved in the type I interferon responses of these cells. To this end, we analyzed exomes, transcriptomes, proteomes, and RNA editing by ADAR enzymes, the latter being found to modulate interferon responses [15]. Some hypotheses resulting from this multiomic data integration were tested in functional assays modeling VSV infection.

## 2. Results and Discussion

### 2.1. Type I Interferon Treatment Protects the DBTRG-05MG Cell Line But Does Not Prevent the HOS Cell Line from VSV Infection

The selection of the cell models for the study was based on the apparent differences in the cells’ response to VSV infection after the interferon treatment, which indicates the differences in the respective interferon-regulated pathways. Two cell lines, glioma-derived DBTRG−05MG and osteosarcoma-derived HOS, both sensitive to VSV, were selected. After 24 h of treatment by interferon α2β, these cell lines reacted differentially to the same viral load. Glioblastoma cells were protected by the type I interferon treatment, whereas osteosarcoma cells were still sensitive with a minimal decrease in cell lysis (Figure 1). These experiments model responses to OV therapy, where HOS represents a tumor prone to the therapy, while DBTRG-05MG mimics a resistant tumor. Thus, the multiomic studies were performed in the follow-up efforts to find specific biomarkers responsible for the observed differential response.

### 2.2. Exome Analysis of DBTRG-05MG and HOS Cell Lines Confirmed Their Authenticity

Exomes of both cell lines were sequenced before, according to their accessions in the Cellosaurus knowledgebase [16]. Then, the particular cell lines at hand were re-sequenced (Figure 2). SNP calling made for our exome data confirmed identities for the cell line of interest and identified some of the drivers making these cells cancerous. Specifically, there was a missense mutation within the TP53 tumor-suppressor gene in HOS. In addition, DBTRG−05MG exhibited actionable V600E mutation in the BRAF receptor kinase gene, frequently found across different types of cancers [17].

### 2.3. A-to-I RNA Editing in DBTRG-05MG and HOS Cell Lines before and after Type I Interferon Treatment

Adenosine-to-inosine RNA editing is one of the frequently observed types of post-transcriptional modifications in animals [18]. In human cells, it is catalyzed by two isoforms of RNA-dependent adenosine deaminases (ADAR). The human ADAR1 enzyme, encoded by the ADAR gene, is responsible for transcriptome-wide editing of double-stranded RNA. In contrast, ADAR2, encoded by the ADARB1 gene, edits only some mRNAs, which may modify protein structures. In addition, ADAR1 has a type I interferon-inducible splice isoform, p150, in contrast to the constitutive shorter p110 isoform, which has been shown to attenuate the interferon response by reducing cytoplasmic levels of dsRNA [15]. Accordingly, mutations in the human ADAR gene lead to upregulated interferon signaling and systemic inflammatory responses; they have been classified as casual for the Aicardi-Goutieres hereditary syndrome [19]. Thus, we hypothesized that increased activity of ADARs, especially ADAR1, may downregulate the type I interferon response in HOS cells and be responsible for the preserved sensitivity to VSV after interferon treatment. However, quantitative data on the expression of ADARs and some functionally associated genes in the considered cell lines did not confirm this assumption (Figure 3). As previously described, both cell lines demonstrated induction of the more extended p150 splice variant of ADAR1 after interferon treatment (Figure 3B). However, in DBTRG-05MG cells, baseline levels of ADAR (Figure 3A) and ADARB1 gene products were significantly higher than in HOS cells (Figure 3B). Notably, at the same time, the inactive ADARB2 (ADAR3), which acts as an inhibitor of RNA editing activities of the ADAR family [20], was expressed only in HOS cells, similar to another ADAR inhibitor, AIMP2 [21]. Finally, the expression of another putative regulator of ADARs, SRSF9 [22], did not differ significantly between the two cell lines under study.

In addition to quantitative gene expression study for relevant products, the indices of A-to-I RNA editing were deduced from transcriptomic data using traces of expressed Alu repeats as a recognized subject of this editing type [23]. Again, the baseline index was much higher for DBTRG-05MG than for HOS, proportionally increased after interferon treatment (Figure 4). Notably, Alu RNA editing index calculations were in good agreement with gene expression data.

These results imply that the expression levels of ADAR family genes and this type of RNA editing activity do not confer compromised type I interferon antiviral activity in HOS osteosarcoma cells. Instead, the more pronounced quantitative and functional activity of ADARs in DBTRG-05MG cells may be due to their tissue origin. It is known that A-to-I RNA editing is generally more active in brain tissues than in connective tissues [18].

### 2.4. Combining Transcriptomic and Proteomic Data for Type I Interferon Response of DBTRG-05MG and HOS Cells Helps to Define Differentially Expressed Genes

DBTRG-05MG and HOS cells were treated with interferon α2β at 1000 U/mL for 24 h, and responses were measured by transcriptome and proteome assays. Both cell lines showed significant changes across many differentially expressed proteins. Since comparison of interferon responses was the focus of this study, it was necessary to subtract the list of differentially expressed proteins for the osteosarcoma cells from the corresponding list for the glioma cells. Indeed, these two lists could explain the difference between cells in their sensitivity to VSV after interferon treatment. Unexpectedly, we encountered difficulty in defining a comparable scale difference in gene expression in both cell lines. Averaged gene expression in DBTRG-05MG cells was less dispersed between replicates in comparison with HOS cells, with the coefficient of variance for the former being 5.8% and 4.9% before and after interferon treatment, and 10.1%, and 8.2%, for the latter, respectively. The treatment proportionately decreased variance, which was explained by the addition of many up-or down-regulated proteins due to the action of cytokines. In triplicate, genes with a similar fold change in expression in two cell lines would have different *p*-values. That would lead to a skewed estimate of the number of differentially expressed genes for the two cell lines if we based our inclusion criteria on these metrics. Thus, a different way of considering the cut-off of the differential expression was used, as described below.

Studying the correlation between RNA and protein abundances is of interest in proteogenomics, often resulting in ambiguous or contradictory results [24]. Despite the apparent causal relationship between gene expression and protein production, there are numerous observations of low correlation [25,26]. A popular explanation for the low correlation is the different lifespans for mRNAs and protein products, with the correlation increasing when multiple time points are included in the study [25]. To further explore the latter idea, for two cell models considered in this work, mRNA-seq-derived data for gene expression and label-free protein quantitation data at different fold cut-offs of transcript level changes after interferon treatment were analyzed (Figure 5A). The figure shows that the correlation for all genes is relatively low (about 0.3) because many genes that do not respond to interferon oscillate with their products at both the transcript and protein levels. An increase in the fold change cut-off leads to a sharp rise in the transcript-protein correlation since the share of co-regulated proteins also increased in the list. With a further increase in the cut-off, the absolute number of genes involved falls significantly, decreasing the correlation (Figure 5A). The fold level of change corresponding to a correlation maximum of about 1.4 was then used as a cut-off to define the differentially expressed genes in the transcriptome data for both cell lines. The difference on the right side of the plot can be explained by the fact that the DBTGR-MG cell line demonstrated more genes undergoing a substantial fold change after type I interferon treatment, confirming the observation of better resistance of these cells to VSV, as described above (Figure 1).

### 2.5. Differential Transcriptomics of Type I Interferon Responses in the VSV Protected DBTRG-05MG and VSV Sensitive HOS Cell Lines

After empirically determining the fold change cut-off for differentially expressed genes in the above cell models, it becomes possible to deduce the gene products responsible for the VSV resistance after type I interferon treatment. To this end, the transcriptomic data were more representative in terms of the numbers of differentially expressed products than the proteomic data. At the 1.4-fold change cut-off, the differentially expressed genes after interferon treatment were 442 and 963 for HOS and DBTRG-05MG, respectively, with 236 in common (54% and 25%, respectively). Much more genes responded to interferon treatment in DBTRG-05MG cells, reflecting the more pronounced response of this cell line.

The differentially expressed genes in each cell line were then analyzed for molecular pathway enrichment [27]. First, enrichment was calculated independently without considering the genes regulated by the treatment in both cell lines (Table 1). The RNA editing data above showed that both cell lines were responsive to type I interferon. On the 10 pathways enriched in differentially expressed genes in VSV-protected DBTRG-05MG cells, seven were also over-presented in virus-sensitive HOS cells. The interferon response pathways top this list with the highest degree of enrichment. Notably, the induction of three pathways was not characteristic of the HOS cell line. The first is TNFα signaling through NFkB, a well-recognized pathway that is closely connected and overlaps with other inflammatory response pathways. Less expected, early and late estrogen response genes were upregulated by type I interferon, specifically in DBTRG-05MG cells. In earlier studies, estrogens, particularly estradiol, induced the production of type I interferon [28]. Our results showed an inverse relationship between the estrogen and interferon pathways: the effects of this cytokine mimic those of estrogens.

To identify factors that specifically confer more profound protection against VSV in the studied glioma cells, we performed differentially expressed gene list subtraction and calculated pathway enrichments for 727 genes that were specifically differentially expressed in the DBTRG-05MG cell line (Table 1, column 2). Again, the inflammatory response pathway was most enriched with other closely related and overlapping pathways, such as TNFα signaling through NFkB and IL6-Jak-Stat3. As for genes with expression specifically induced in DBTRG-05MG and related to the inflammatory response, they are listed and categorized in Table 2. These are secreted proteins, including CXCL family cytokines, TNF-like factors, EREG (the EGF-like regulator), etc. Secreted factors and cell adhesion molecules are involved in the recruitment of various subclasses of immune cells stimulated by type I interferon. However, the effects rendered by these proteins are not relevant to cultured cells. Therefore, the study aimed to identify specific differences between cells that respond and do not respond to treatment with interferon. Thus, this knowledge does not allow monitoring or predicting resistance to VSV after type I interferon treatment. At the same time, these results are consistent with previous work, in which two hepatoblastoma cell lines responded differentially to interferon treatment with respect to hepatitis C virus replication [11]. The differential expression of genes encoding secreted cytokines, such as CXCL10, CXCL11, and EREG, indicated antiviral resistance after interferon treatment. Notably, the same genes showed similar effects in the cell lines used in the present study and marked resistance to another single-stranded RNA virus, VSV (Table 2).

Transcriptomic data demonstrated the similarity of interferon-activated pathways in both cell lines. However, the overall response to the interferon treatment, as determined by the number of differentially expressed genes, was more pronounced in the VSV-protected glioma cells. Furthermore, this response was enriched with secreted cytokines that function in vivo as attractors of immune cells, and this effect was hardly relevant to the environment in cell culture. Therefore, it can be assumed that, in the DBTRG-05MG cells, the molecules that protect VSV reached certain quantitative, stoichiometric thresholds of the corresponding pathways. Yet, the pathway enrichment analysis has not suggested possible specific biomarkers for the existence of this defense or targets for its management. However, we could further scrutinize the list of differentially expressed proteins to tell which ones could be for their intended use.

### 2.6. EGF Receptors in DBTRG-05MG and HOS Cell Lines

Considering differences between the cell lines in the interferon-induced signaling pathways that can be regulated experimentally, we first assessed the receptors for secreted interferon-induced cytokines in the glioma cell line only (Table 2). Among them, the most promising results showed to express EREG EGF-like factors, *EGFR* and *ERBB2*, which encode HER2, a co-receptor for EGFR, a well-known anti-cancer therapeutic target. The genes for both EGF receptors demonstrated significantly higher expression levels in the HOS cell line, which was sensitive to VSV after type I interferon treatment (Figure 6A,B). Also, these receptors are not known to be regulated by interferon, and, as would be expected, their expression levels did not change after the treatment. Recent reports show a cross-talk between EGF and type I interferon signaling [30]. In particular, HER2, after its internalization, was able to bind STING, the primary cytoplasmic DNA sensor, and thus an inductor of interferon production through the IRF3 transcription factor. HER2 binding to STING leads to phosphorylation of another STING interactor, TBK1, further attenuating STING signaling and enhancing STING-mediated antiviral immunity [31] (Figure 7). Interestingly, the STING and IRF3 genes were also overexpressed in HOS compared to DBTRG-05MG cells (Figure 6C,D). However, we speculate that the corresponding pathway may be compromised in HOS cells due to the post-translational regulation described above.

In addition, after inhibition of EGF receptors with erlotinib, type I interferon induction was observed in both in cello and in vivo models of lung cancer [32]. This induction has been shown to promote tumor cell survival by mutant EGFR, and therefore interferon receptor inhibitors have been suggested as a therapy for some lung cancers.

**Figure 7 ijms-23-05244-f007:**
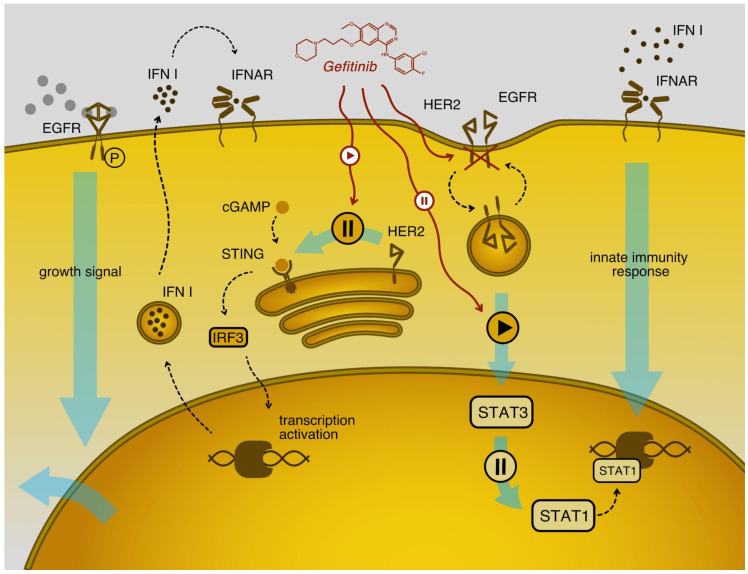
Cross-talk between EGR and type I interferon signaling. The binding of ligands with EGF receptors initiates cell proliferation and suppresses the innate immune response. After binding the ligand, HER2 molecules are embedded in the vesicle by endocytosis and transfer to the Golgi complex. Then they inhibit STING, which is a key stimulator of type I interferon gene transcription via IRF3. Treatment with EGFR inhibitors, such as gefitinib, inhibits HER2 receptors and suppresses the STING activation, further increasing the innate immunity response (a left part of the scheme based on results shown elsewhere [33,34]). In addition to inhibition of type interferon production, EGF receptor signaling may attenuate type I interferon signaling from its receptors, changing the ratio between active forms of STAT1 and STAT3, the latter being considered as inhibiting the main response pathway based on activated STAT1 dimer (right part of the scheme based on results from [30]).

Many recent studies have shown that EGF signaling, namely the activation of HER2 followed by its internalization, can attenuate the production of type I interferon. Accordingly, inhibition of this pathway by erlotinib leads to enhanced interferon induction. However, in this work, where the oncolytic therapy by VSV was modeled with and without type I interferon treatment, the cytokine was administered topically. Thus the effect was not relevant to a systemic induction of type I interferon. Moreover, we found no difference in type I interferon expression levels between the two cell lines studied. Thus, EGF signaling was presumably attenuated downstream of the interferon receptor pathways. Indeed, one of the first works exploring the cross-talk between EGF and type I interferon signaling already observed the synergism between erlotinib and interferon α treatment in resistance to hepatitis C virus (HCV) [30]. Thus, besides stimulating interferon induction by inhibiting the HER2-Akt1-STING axis, as shown in Figure 7, erlotinib and its analogs may also activate the type I interferon signaling from its receptor, more likely, through competition between the STAT1-STAT3 transcription factors [30]. Here, we further explored the hypothesis that EGFR inhibition may increase the type I interferon response in the VSV-sensitive HOS cells and, to some extent, protect it from the virus by acting synergistically with the extrinsic interferon α2β. Accordingly, we hypothesized that HER2 protein levels might indicate the expected sensitivity of cells to VSV after interferon treatment. We used gefitinib as an EGF receptor inhibitor widely marketed to treat EGFR-mutated cancers to test this hypothesis.

### 2.7. Gefitinib Stimulates Type I Interferon Signaling in Cancer Cell Models

To confirm the synergistic effect of gefitinib on type I interferon signaling similar to that found in the earlier work on erlotinib [31], publicly available transcriptomic data were obtained for the cancer models in which the cells were treated by gefitinib were used. Three sets of data on human cancer cells were selected. In the first dataset taken from ArrayExpress [35], A-431 epidermoid skin carcinoma was treated with this drug for 24 h [36]. Two other data sets available on the Gene Expression Omnibus [37] represent the results of more extended treatment with gefitinib for up to 2 weeks, including PC-9 lung adenocarcinoma and MGH119 non-small cell lung carcinoma [38]. Pathway analysis of genes differentially expressed after treatment with gefitinib convincingly demonstrated upregulation of type I interferon signaling by this EGFR inhibitor. Using both Reactome [39] and Gene Ontology [40] pathway analyses, the corresponding pathways were enriched in the first ranks (Table 3). Expectedly, gefitinib action appears to be similar to the one of erlotinib in its ability to enhance type I interferon production and signaling.

### 2.8. Gefitinib Influences the Type I Interferon Response in VSV Sensitive Cell Lines, Enhancing Their Protection from the Virus

The synergy between type I interferon signaling and gefitinib treatment was tested in DBTRG-05MG and HOS cells as models of VSV oncolytic therapy. Also, the U251MG and A172 glioblastoma cell lines [41] were added to the study due to their intermediate behavior between DBTRG-05MG and HOS cells. These cell lines exhibit only partial protection against VSV after interferon treatment. In addition, a four-culture panel of primary glioblastoma cell lines was tested for HER2 expression and its ability to be protected from VSV infection by treatment with interferon, gefitinib, or both. Finally, all cell lines were tested for virus resistance using cytotoxicity and viral replication efficiency parameters after the treatment with interferon α2β, gefitinib, or both.

Gefitinib alone or in combination with interferon α2β has no significant effect on VSV infection in DBTRG-05MG cells and one of the primary glioma cell cultures (prGl2, Figure 8). First, EGF receptors are expressed at low levels in the former cell line, as shown above. Second, interferon alone was shown to protect DBTRG-05MG from VSV (Figure 1); hence, gefitinib did not significantly increase the effect (Figure 8). In contrast, HOS, U251MG, A172 cells, and three of four primary glioma cultures responded to gefitinib treatment with reduced VSV sensitivity and lower replication efficiency (Figure 8). Interestingly, the anti-EGFR drug itself was protective, meaning that after at least 24 h of gefitinib treatment, endogenic interferons could be produced at a level sufficient to enhance the antiviral defense at a lower multiplicity of infection. These experiments showed that gefitinib and its analogs act through endogenous interferon production, as described earlier [32], and through the cross-talk with the interferon signaling [30].

### 2.9. Levels of HER2 Protein Are Higher in Cell Lines with Attenuated Type Interferon Response to VSV Infection

As mentioned before, HER2 co-receptor activation and internalization to Golgi complex vesicles can impair type I interferon signaling [31]. According to this logic, higher levels of this protein in a cell line will indicate a lower response to type I interferon in terms of protection from VSV. These levels were measured by Western blot for all cell lines used in this work, DBTRG-05MG, HOS, U251MG, A172, and four primary glioblastoma cultures. Indeed, in the cell lines with statistically significant responses to gefitinib treatment, HOS, U251MG, and A172, the levels of HER2 were significantly increased compared with non-responding DBTRG-05MG (Figure 9). As for primary cultures, the level of HER2 expressions varies in a wide range. The only line without response to gefitinib treatment, prGl2, had a deficient HER2 expression and preserved the ability to respond to treatment with interferon. These results suggest that the HER2 level in a tumor may be a predictor of response to viral oncolytic therapy. Combining anti-EGFR therapy with interferon-sensitive oncolytic viruses could not be optimal for tumors with higher HER2/neu expression. Proving this suggestion on a much more extensive array of samples, including cell lines and murine models of viral oncolytic therapies, will be a subject of follow-up efforts.

### 2.10. EGF Treatment of Cell Lines Confirmed That the Effect of Gefitinib Is Provided by EGF Signaling

We supposed the effect of gefitinib on cell sensitivity to VSV was due to EGFR signaling inhibition. In that case, it is reasonable to assume that the treatment of cells with EGF may lead to the opposite effect, i.e., increasing the sensitivity of cells to the virus. We found that pretreatment of cells that express a lot of HER2 (U251MG and HOS cells) with EGF at a 50 ng/µL for 24 h resulted in a significant increase in their sensitivity VSV (Appendix A). It can be assumed that EGF cannot only stimulate cell proliferation but also promote more efficient virus replication, for example, by stimulating protein synthesis and, presumably, inhibiting innate immunity.

To rule out that its toxicity caused the effects of gefitinib, virus-mediated cytotoxicity was revealed in our model systems by measuring the metabolic activity of mitochondria and the intracellular level of ATP. First, we treated the cells for 48 h with EGF or gefitinib to rule out possible artifacts. Then, we analyzed the metabolic activity of cells by mitochondrial activity (MTT test) and by measuring intracellular ATP levels (cell titer assay). There was a substantial increase in metabolic activity in the cells treated with EGF and only a slight decrease after gefitinib treatment (Appendix A). The absence of changes in cell morphology argued against any toxicity at the used concentrations of EGF and gefitinib.

A key experiment that would confirm the physical involvement of HER2 in downregulating type I interferon in cells with higher levels of EGF receptor signaling could be the overexpression of this co-receptor in cells lacking this protein and thus protected from VSV by interferon. The overexpression should lead to the partial or complete abolition of this protection. Unfortunately, the DBTRG-05MG cell line, which is the best example of an interferon-protected cell line, appears to be highly resistant to transduction, including DNA transfection and lentiviral transfer. Furthermore, the length of HER2, which consists of ca. 1300 amino acid residues, also provides technical risks for its overexpression. The negative results of the transfection experiments are described in Appendix A. To summarize, we must state that direct proof of HER2 responsibility for antagonistic action towards type I interferon signaling was not technically possible with the cell models used in this work.

## 3. Materials and Methods

### 3.1. Cells and Viruses

Glioblastoma cells (U-251 MG and DBTRG-05MG) and osteosarcoma cells (HOS) were purchased from ATCC and maintained in DMEM medium with 4.5 g/L glucose, 10% bovine serum (BioSera, France), and 2 mM glutamine (PanEco, Russia). In addition, VSV (Indiana strain), kindly gifted by Oleg Zhirnov (D.I. Ivanovsky Institute of Virology, a division of N.F. Gamaleya National Research Center of Epidemiology and Microbiology, Moscow, Russia), was used for virus infections.

For primary cultures obtaining tumor, fragments were disintegrated mechanically with a scalpel and washed with phosphate-buffered saline within 24 h after excision. Then further homogenization of the tumor was carried out using a strainer with a pore diameter of 200 μm. The resulting suspension was sown on culture plates with DMEM medium (PanEco, Russia) with 10% fetal bovine serum (Gibco, USA) and the addition of an antibiotic and antimycotic (Anti-anti, Invitrogen, Waltham, MA, USA). Then, for the next 14 days, the medium was changed every 2 days; the cells attached to the plastic formed colonies. Finally, the cells were passaged using standard methods: they were detached by washing with a versene solution, then treated with trypsin 0.025% (PanEco, Russia). In the experiments, 15–25 passages of cells were used, the morphology of which remained unchanged during the entire study.

### 3.2. Cell Cultures for Multi-Omics Analysis

DBTRG-05MG and HOS cells were grown in a DMEM medium (PanEco, Moscow, Russia) supplemented with 10% fetal bovine serum (Gibco, Thermo Fisher Scientific, Waltham, MA, USA). The cell cultures were treated with interferon (IFN) α2β at a concentration of 100 units/mL for 24 h, and the untreated control were subjected to further analysis. The sub-confluent cell cultures were grown and treated in 6 cm culture plates. The cells were scraped from the surface, washed three times with cold PBS, and pelleted by low-speed centrifugation for further use. Cell preparations were made in independent triplicates.

### 3.3. Interferon and Gefitinib Treatment and Infection

To test the sensitivity of the cell line to interferon treatment and the effectivity of interferon response, cells were plated on 96-well plates (3000 cells per well). The next day, cells were treated with recombinant IFN-α2β (Pharmapark, Moscow, Russia) at 1000 U/mL and/or gefitinib at a concentration 2 µM. Human EGF recombinant protein (PanEco, Russia) was used in 12, 25, and 50 ng/mL concentrations. After 24 h, cells were infected with VSV in a wide range of MOI (100–0.001) in a DMEM medium. One hour after viral adsorption, the medium was changed to DMEM with 1% FBS. After 48 h, cell viability was assessed using the CellTiter-Glo^®^ Luminescent Cell Viability Assay (Promega, Madison, WI, USA) according to the manufacturer’s protocol. Reed and Muench’s method was used to obtain the TCID50 value [42]. Three independent biological replicates were analyzed in quadruplicates.

To test replication efficiency, cells were seeded on a 12-well plate. Then, 50% confluent monolayers were treated with recombinant IFN-α2β (Pharmapark, Moscow, Russia) in a concentration of 1000 U/mL and/or gefitinib in a concentration of 2 µM. In 24 h, cells were infected by incubating VSV with cells for 1 h (MOI = 0.1) at 37 °C in a DMEM medium. Media with the virus was aspirated from cells, followed by the addition of fresh media. In 24 h after infection, supernatants were harvested. Replication efficiency was determined by infection of the BHK21 cell line with serial dilutions of supernatants (Reed and Muench method).

### 3.4. Western Blotting

Cells were cultured on 60 mm Petri dishes with confluency of 50%, washed with cold PBS, and lysed in 600 µL of RIPA Lysis and Extraction Buffer (ThermoFisher Scientific, Waltham, MA, USA) containing a Roche protease inhibitor cocktail (Sigma-Aldrich, St. Louis, MO, USA), according to the manufacturer’s instructions. Lysates were centrifuged for 15 min at 12,000× *g*; the Bradford assay measured the total protein content in the protein supernatant. Proteins were separated by electrophoresis in a 12% SDS-polyacrylamide gel and transferred to a PVDF membrane (Amersham Hybond P 0.45 µm, Amersham Biosciences, GE Healthcare, Chicago, IL, USA). Membranes were blocked in a 4% solution of no-fat milk in PBST (PBS with 0.05% Tween 20) for 1 h at room temperature. As primary antibodies, anti-beta-actin sc-47778 diluted at 1:3500 (Santa Cruz Biotechnology, Santa Cruz, CA, USA) and ErbB2/Her2 Antibody AF1129 diluted at 1:3000 (R&D Systems, Minneapolis, MN, USA) were used for actin control and HER2 detection, respectively. As secondary antibodies, m-IgGκ BP-HRP, sc-516102, diluted 1:3000, and mouse anti-goat IgG-HRP, sc-2254, diluted 1:2500 (both Santa Cruz Biotechnology, Santa Cruz, CA, USA) were used for beta-actin and HER2, respectively. Incubation with primary and secondary antibodies was performed overnight at 4 °C and for 1 h at room temperature. Clarity Western ECL Substrate was used for chemiluminescence detection using the Bio-Rad ChemiDoc MP Imaging System (Bio-Rad Laboratories, Berkeley, CA, USA). Protein bands were quantified using ImageJ software (NIH, Bethesda, MD, USA). Data were depicted as bar graphs indicating the mean and SD as fold change compared to the control. For each comparison, a *t*-test was used (GraphPad Software, San Diego, CA, USA). Uncropped scans of the blots are provided in Appendix A.

### 3.5. Nucleic Acid Isolation

For nucleic acid sequencing, 1.5 × 10^6^ cells of each line, treated with IFN α2β and untreated control, were taken from three independently defrosted cultures. DNA was extracted using Gentra Puregene Cell Kit (Qiagen, Hilden, Germany) according to the manufacturer’s instructions. Total RNA was extracted using an RNeasy mini kit (Qiagen, Hilden, Germany). For real-time qPCR analysis, total RNA from 1 × 10^6^ cells was extracted using an RNeasy mini kit (Qiagen, Hilden, Germany) in four biological replicates and reverse transcribed using an MMLV RT kit (Evrogen, Russia).

### 3.6. Real-Time PCR for ADARs and Functionally Related Genes

Real-time PCR was performed with SYBR Green qPCR master mix qPCR mix-HS SYBR (Evrogen, Russia) in a Rotor-Gene Q (Qiagen, Germany) detection system. The PCR protocol was as follows: an initial activation at 95 °C for 3 min, 39 cycles at 94 °C for 15 s, 61 °C for 10 s, and 72 °C for 15 s. Ct values were converted into relative gene expression levels compared to internal control genes, β-actin, and TBP. Each PCR run was performed in triplicate. The primer sequences were as follows: ADAR_p150-F, AATGGATGGGTGTAGTATCCGC; ADAR_p150-R, CGGGCAATGCCTCGC; ADAR-F, GTAGATCCCTGCGGTAACGG; ADAR-R, AGGAGACAAGCGTCAACTGG; ADARB1-F, CGGTCAGGTCACCAAACTTACC; ADARB1-R, CCGCAGGTTTTAGCTGACG; ADARB2-F, ACGATGCTCTGCAGGTACAC; ADARB2-R, CAAGATCGAGTCCGGGGAAG; AIMP2-F, GTTTTCAGGCACGCTCTTG; AIMP2-R, AGTGCTTGGGAAGGATTACG; SRSF9-F, GAACTCCACACGAAGCCGAC; SRSF9-R, GATCGAGCTCAAGAACCGGC; β-actin-F, GTCTCAAACATGATCTGGGTC; β-actin-R, CACCACACCTTCTACAATGAG; TBP-F, TCTGGGTTTGATCATTCTGTAG; TBP-R, GAGCTGTGATGTGAAGTTTCC. Primers were purchased from Evrogen (Moscow, Russia).

### 3.7. Library Construction and High-Throughput DNA and RNA Sequencing

For RNA-seq, PolyA transcriptomic libraries were constructed using MGIEasy RNA Library Prep Set (BGI, Beijing, China) according to the manufacturer’s protocol. First, genomic DNA was randomly fragmented by Covaris. Then exome libraries were made using Agilent SureSelect Human All Exon V6 kit (Agilent, Santa Clara, CA, USA). All libraries were sequenced on DNBSEQ-G400 (BGI, Beijing, China), resulting in paired-end reads of 100 bp. The read quality was examined with the FastQC program (Babraham Bioinformatics, Cambridge, UK). The sequencing was done as an external service from Genomed (Moscow, Russia).

### 3.8. Alu Editing Index Quantification

RNA-Seq reads were mapped to the human genome (version hg19) using STAR [43] with options—outFilterMatchNminOverLread 0.95—outSAMmultNmax 1 to obtain the 95% minimal rate of mismatches and exclude multi mapping reads. After that, Alu editing index (AEI) was calculated using the AEI tool developed by Roth et al. [23]. The Student’s *t*-test evaluated the statistical significance of the difference between the obtained values.

### 3.9. Differential Expression Analysis

To discover the differentially expressed genes, reads were mapped to the human genome (version hg19) using STAR with the option “—outFilterIntronMotifs RemoveNoncanonical” to remove the non-canonical splice junctions. Option “—quantMode GeneCounts” counted the number of reads overlapping annotated regions. The human genome annotation was obtained from GENCODE (Release 19, GRCh37.p13). The identification of differentially expressed genes was carried out using Bioconductor package edgeR [44].

### 3.10. Single Nucleotide Variation Calling

The SNV calling was performed according to the GATK best practices protocol [45]. Briefly, exome reads were mapped to the human genome (GRCh37, GENCODE release_30) using the BWA aligner [46]. Picard tools were used to mark duplicates [47]. Further steps included base quality score recalibration, haplotype caller, and hard filtering of variants with options recommended within the best practices pipeline.

### 3.11. Proteomics Data Collecting

Following LC-MS/MS analysis, sample preparation was performed as described in [10]. Briefly, cells were resuspended in 100 μL of lysis buffer (0.1% *w*/*v* ProteaseMAX Surfactant (Promega) in 50 mM ammonium bicarbonate and 10% *v*/*v* ACN) and sonicated for 5 min at 30% amplitude on ice (Bandelin Sonopuls HD2070, Bandelin Electronic, Berlin, Germany). Protein extracts were reduced in 10 mM dithiothreitol at 56 °C for 20 min, alkylated in 10 mM iodoacetamide at room temperature for 30 min in the dark, and digested at 37·°C overnight with trypsin (Sequencing Grade Modified Trypsin, Promega). Trypsin was deactivated by adding acetic acid (5% *w*/*v*). Samples were desalted using Oasis cartridges (Oasis HLB, 1 cc, 10 mg, 30 μm particle size, Waters) and loaded at 1 μg per injection. Data were collected in data-dependent acquisition mode using Orbitrap Fusion Lumos mass spectrometer (Thermo Scientific, Waltham, MA, USA) coupled with UltiMate 3000 nanoflow LC system (Thermo Scientific). Analytical column EASY-Spray PepMap RSLC C18 (2 μm, 75 μm i.d. × 500 mm, 100 Å) (Thermo Scientific) was employed for linear gradients from 5% of B to 20% of B for 105 min, followed by a linear gradient to 32% B for 15 min at 270 nL/min flow rate. Mobile phases were as follows: (A) 0.1% formic acid (FA) in water; and (B) 95% acetonitrile, 0.1% FA in water. Precursor ions were measured in mass range *m*/*z* from 375 *m*/*z* to 1500 *m*/*z* with resolving power of 120,000 at *m*/*z* 200, maximum injection time of 50 ms, and automatic gain control (AGC) of 4 × 10^5^. Ion isolation in the *m*/*z* window of 0.7 Th followed fragmentation using higher-energy collision dissociation (HCD) at normalized collision energy (NCE) of 30%. Fragment ions were measured in the Orbitrap mass analyzer with a resolving power of 30,000 at *m*/*z* 200.

### 3.12. Proteomic Data Processing

Raw files were converted into mzML format using msConvert from ProteoWizard (v. 3.0.20066). Database search was performed using IdentiPy (v. 0.2) [48]. Parameters for the search were 5 ppm precursor mass accuracy, 0.01 fragment mass accuracy, and up to two missed cleavage sites. Carbamidomethylation of cysteine was used as fixed modification; oxidation of methionine and N-terminal formylation were used as variable modifications. Peptide MS1 intensities were extracted by IdentiPy using Dinosaur (v. 1.2.0) software [49]. Label-free protein quantitation was performed using Diffacto software [50].

### 3.13. Data Availability

Genomic and transcriptomic data are available from Gene Expression Omnibus [37] project accession GSE166877. Proteomic data are available from the Proteomexchange repositorium [51], project accession PXD022868.

The differential expression analysis of transcriptomes for DBTRG-05MG and HOS cell lines after type I interferon treatment is provided in Appendix A, respectively. The results of label-free quantitation analysis of proteomes for the same condition and the same cell lines are provided in Appendix A, respectively.

## 4. Conclusions

In this study, we aimed to scrutinize the type I interferon response to protect cancer cell lines from lysis by the vesicular stomatitis virus for the oncolytic virus therapy models. Two polar responses were considered, including complete protection in the case of the DBTRG-05MG glioma cell line and a lack of such protection in the case of the HOS osteosarcoma cell line. In addition, the molecular response to external treatment by interferon α2β was studied in those cells at transcriptome, proteome, and RNA editome levels. Comparing these responses and their differences between the two cell lines identifies RNA and/or proteins that can be markers of response prediction and, further used, potentially, in developing precision oncolytic viral therapy.

The most valuable and interpretable results were derived from transcriptomic analysis. Specifically, the inflammatory signaling was enhanced in the responding cell line, manifested in the overexpression of secreted inflammatory cytokines, such as CXC-motif ligands, TNF analogs, etc., and receptors for similar molecules. In addition, the overlapping set of differentially expressed genes was shown in earlier experiments with cell lines, sensitive or resistant to hepatitis C virus [11]. Although these effects can be relevant to the in vivo tumor environment, where such an enhanced response provides a more extensive antiviral protection by recruiting immunocompetent cells, it has nothing to do with the effects observed here for VSV infection modeled in cello.

The results of this study and further analysis of differentially expressed products led us to the EGF receptors, two of which were represented at much higher levels in the sensitive HOS cells. Recent studies have argued for an antagonistic cross-talk between the EGF signal transduction pathway and type I interferon response in cancer cells emerging through the activity of HER2, a co-receptor of EGFR and an important drug target [30,32]. Indeed, an addition of the EGFR inhibitor, gefitinib, alone or simultaneously with interferon treatment, performed in our work, demonstrated developing resistance to VSV for two otherwise sensitive standard cell lines and three primary glioma cultures. Also, we found that HER2 protein was generally overexpressed on sensitive cell lines in contrast to the resistant ones. We further suggest that this protein can potentially be a biomarker of tumor vulnerability to oncolytic therapy by VSV. Unfortunately, overexpression of this protein in cell lines of interest, which could provide direct proof of its physical antagonism with type I interferon response against VSV, was not technically feasible, at least in cell models used in this work. Still, its participation is suggested based on a series of indirect signs we observed in experiments. Regardless, the ability of gefitinib to increase interferon protection of glioma cells from the virus was convincingly proven. These gefitinib effects led us to the clinically significant conclusion that the combination of anti-EGFR therapy with interferon-sensitive oncolytic viruses is not effective for tumors with higher HER2/neu expression. More studies are needed to confirm our findings and suggestions in the follow-up efforts on larger sets of tumor samples.

## Figures and Tables

**Figure 1 ijms-23-05244-f001:**
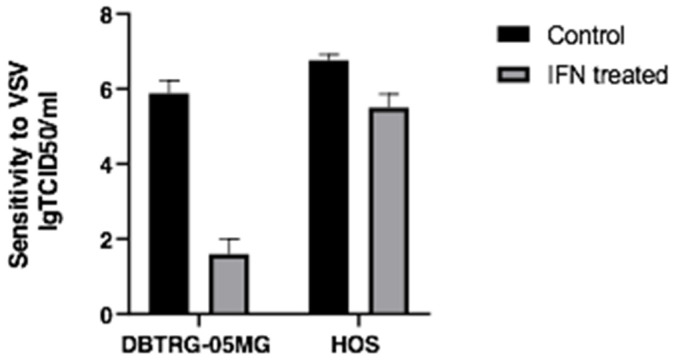
Sensitivity of DBTRG-05MG and HOS cell lines to vesicular stomatitis virus (VSV) with and without interferon α2β pretreatment (24 h, 1000 U/mL).

**Figure 2 ijms-23-05244-f002:**
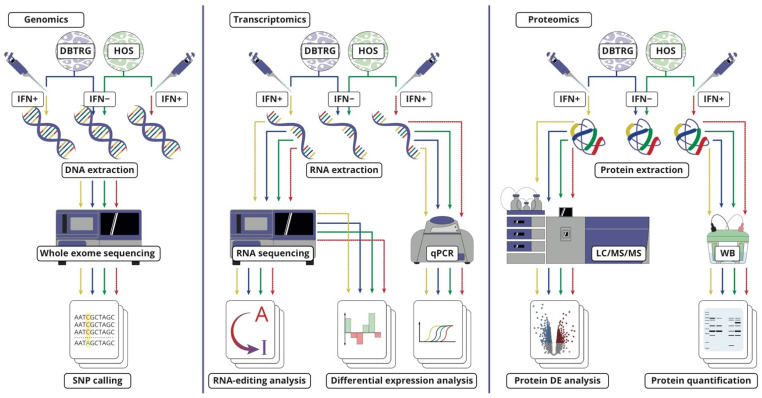
Multiomic workflow of this study. Two cell lines, DBTRG−05MG and HOS, were analyzed at the genomic, transcriptomic, and proteomic levels before and after type I interferon treatment, which protected the former cell line from VSV infection and failed to protect the other one. RNA editing was studied separately by measuring expression levels of relevant genes and a specific bioinformatic analysis of transcriptome data.

**Figure 3 ijms-23-05244-f003:**
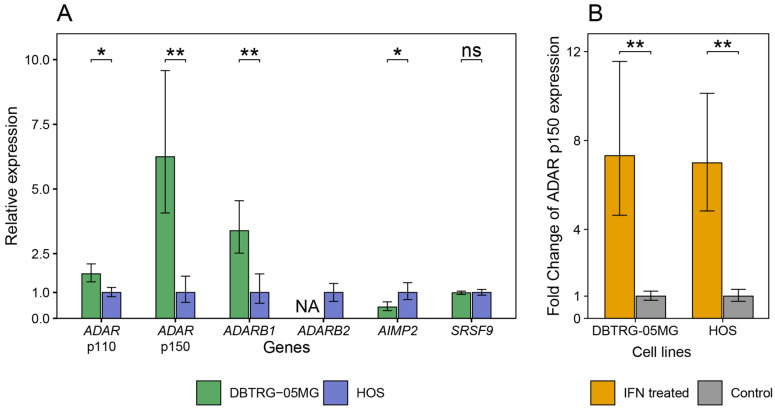
Gene expression of enzymatically active ADAR isoforms and functionally related products in DBTRG-05MG and HOS cells before and after treatment by an interferon α2β. ADAR gene products are represented by short nuclear p110 and long cytoplasmic p150 splice isoforms, both encoding ADAR1 enzymes. ADARB1 encodes a second enzymatically active protein of the family, ADAR2. ADARB2 encodes an enzymatically inactive, possibly inhibitory ADAR3 protein [18]. AIMP2 [21] and SRSF9 [22] products downregulate ADAR RNA editing in some human tissues. In the figure: (**A**) Baseline gene expression levels in cell lines of interest before the treatment; and (**B**) a range of the ADAR p150 splice isoform induction after the interferon treatment. Other products of interest did not change their expression upon treatment (not shown). All measurements were performed in four biological repeats with three technical replicates. Bar heights display the sample mean, and whiskers span the 95% confidence interval between biological replicates. * *p* < 0.05, ** *p* < 0.01, *t*-test.

**Figure 4 ijms-23-05244-f004:**
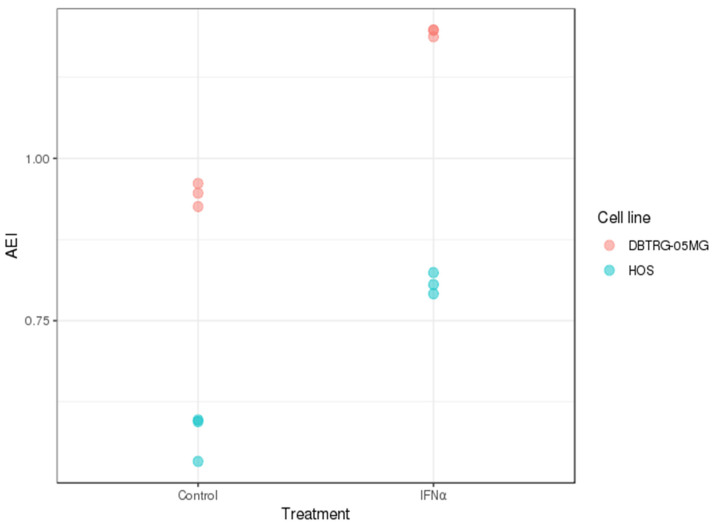
Alu RNA editing indices (AEI) [23] for DBTRG-05MG and HOS cells before and after interferon α2β treatment, which are intended to be surrogate markers for an enzymatic function of ADAR1. Despite the higher basic levels of ADAR activity in the glioma cell line, both cell lines demonstrated approximately equal induction of this activity under treatment. Each state on the graph is represented by data from three biological replicates of the transcriptome.

**Figure 5 ijms-23-05244-f005:**
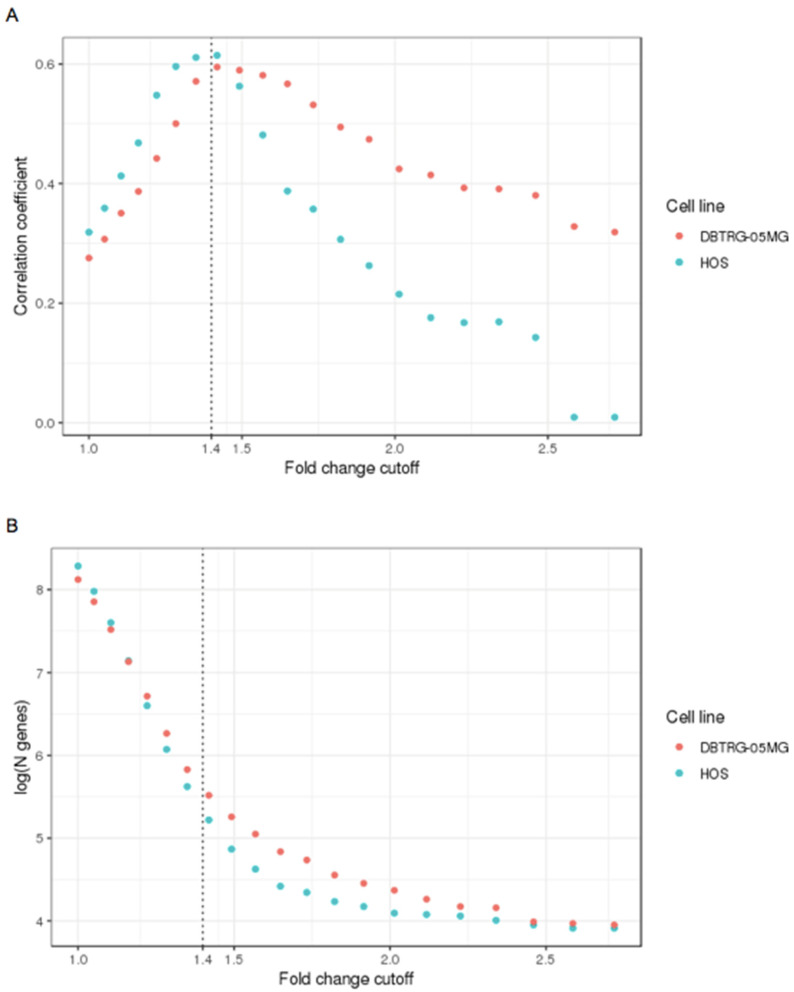
A graphical demonstration of the approach to determining the fold-change cut-off for differentially expressed proteins in DBTRG-05MG and HOS cell lines under type I interferon treatment. Pearson correlation between logFC values obtained by RNA-seq and label-free quantitative proteomics when using gradually varying cut-offs for RNA-seq fold change values (**A**). Amounts of genes are differentially expressed in HOS and DBTRG-05MG cell lines depending on RNA-seq fold change cut-off (**B**).

**Figure 6 ijms-23-05244-f006:**
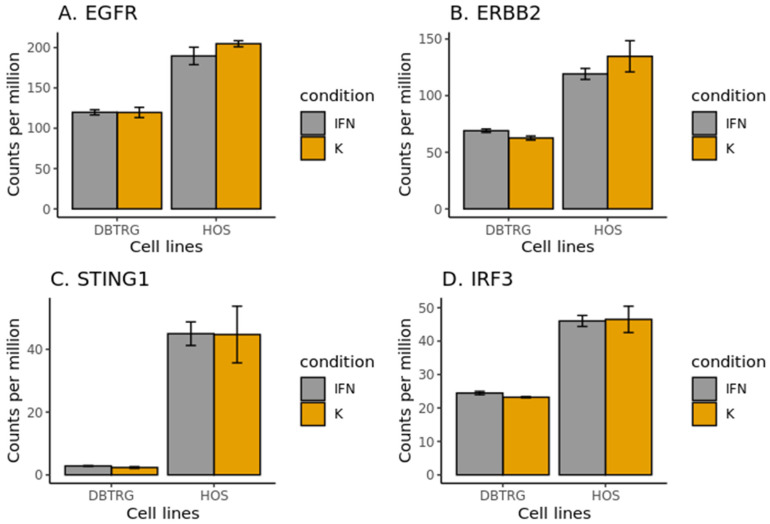
Expression levels of EGF receptor genes of interest and genes involved in type I interferon-inducing pathway inhibited by HER2, in DBTRG-05MG and HOS cell lines, before and after interferon α2β treatment, as determined by RNA-seq. Expression levels of the main EGF receptor genes, *EGFR* (**A**); expression levels of *ERBB2* encoding HER2 co-receptor (**B**); expression levels of *STING1* encoding a major sensor of cytoplasmic DNA and an inductor of type I interferon production, which HER2 inactivates through Akt1 kinase [31] (**C**); expression levels of *IRF3* encoding a transcription factor that induces expression of type I interferon genes and lies downstream of STING in the pathway (**D**).

**Figure 8 ijms-23-05244-f008:**
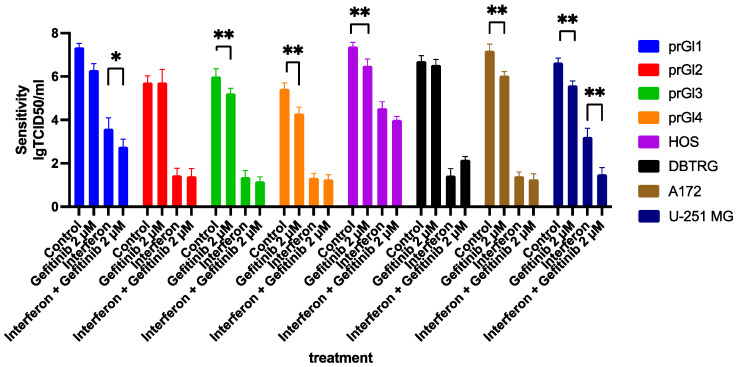
Replication efficiency of VSV on standard and primary glioma (prGl) cell lines, treated with gefitinib (2 µM), interferon α2β (1000 U/mL), and simultaneous treatment of gefitinib and interferon, each for 24 h. Gefitinib alone inhibited viral production in all cell lines except DBTRG-05MG and prGl2. Gefitinib treatment has shown a synergic effect with interferon treatment, inhibiting viral production in prGl1 and U-251 cells. (*n* = 3 replicates per group). Bars represent mean values with SD, * *p* < 0.05, ** *p* < 0.01, *t*-test.

**Figure 9 ijms-23-05244-f009:**
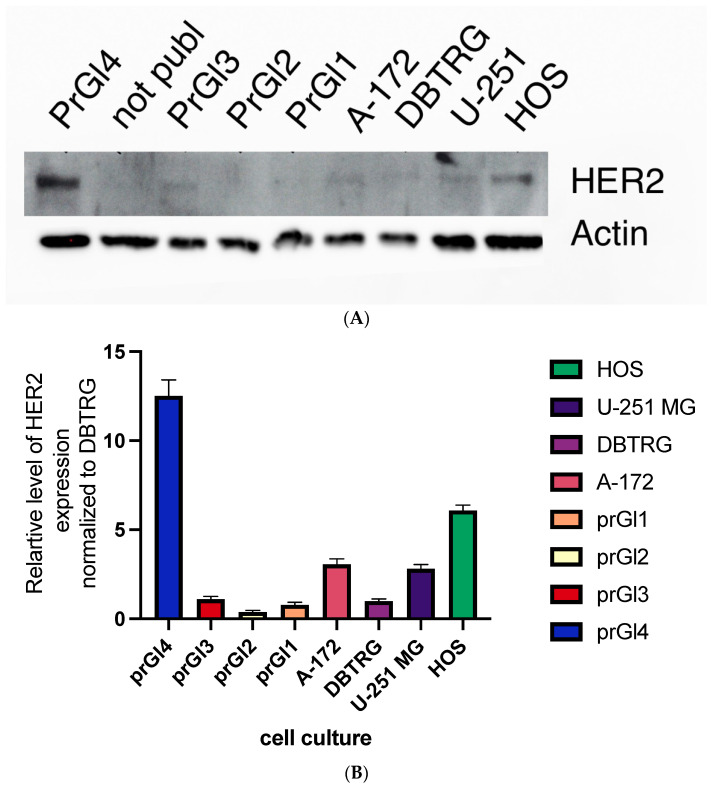
Western blot analysis of HOS, U-251 MG, and DBTRG-05MG cells. The latter developed a complete response to type I interferon as a protection from the subsequent VSV infection. Two other cell lines, osteosarcoma and glioma/astrocytoma, were not completely protected from the virus by interferon treatment. Membranes were incubated with antibodies against HER2 and β-actin for standardization (**A**). The relative ratio of protein expression of β-actin-standardized HER2 levels compared to DBTRG-05MG cells (**B**). Uncropped Western blot scan can be found in Appendix A.

**Table 1 ijms-23-05244-t001:** Hallmark molecular pathways as composed by Molecular Signatures Database [27] enriched in differentially expressed genes after interferon α2β treatment of DBTRG-05MG and HOS cell lines.

Hallmark Pathway	DBTRG-05MG Cell Line, Enrichment *p*-Value for All Differentially Expressed Genes, *n* = 963 *	DBTRG-05MG Cell Line, Enrichment *p*-Value for Genes Differentially Explicitly Expressed in This Cell Line, *n* = 727	HOS Cell Line, Enrichment *p*-Value for All Differentially Expressed Genes, *n* = 442
Interferon-alpha response	10^−84^	Not enriched specifically	10^−99^
Interferon-gamma response	10^−75^	0.008	10^−87^
Inflammatory response	10^−11^	10^−4^	10^−5^
Allograft rejection	10^−8^	0.01	10^−6^
IL6-Jak-Stat3 signaling	10^−6^	0.007	10^−4^
Complement	10^−5^	Not enriched specifically	10^−6^
TNFα signaling via NFkB	10^−4^	0.01	Not enriched
Apoptosis	0.003	Not enriched specifically	10^−4^
Estrogen response, early	0.003	0.007	Not enriched
Estrogen response, late	0.01	0.007	Not enriched

* All *p*-values calculated with multiple comparison correction by the Benjamini–Hochberg FDR method.

**Table 2 ijms-23-05244-t002:** Genes differentially expressed uniquely in the DBTRG-05MG cell line after interferon α2β treatment and attributed to the hallmark inflammatory response pathway [27], which is most enriched in this group of genes. Functional groups are defined by biocuration based on annotations of GeneCards [29].

Gene Names	Functional Group
CXCL6, CXCL10 *, CXCL11 *, TNFSF10, TNFSF15, TNFAIP6, EREG, SPHK1	Secreted regulators of inflammatory response
APLNR, IL1R1 *, TNFRSF9,TNFRSF1B *	Hormone and cytokine receptors
ICAM1, ITGA5, SELL	Immune cell adhesion molecules
NLRP3, CYBB, TAPBP	Pathogen response
BTG2, RNF144B	Antiproliferative and/or proapoptotic
RTP4	Chaperone for G-protein coupled receptors

* Genes also participate in another enriched hallmark pathway, IL6-Jak-Stat3 signaling.

**Table 3 ijms-23-05244-t003:** Type I interferon signaling pathway is enriched by differentially expressed genes in available RNA seq datasets obtained for cancer cell lines treated with gefitinib [36,38]. Pathways were generated by Reactome [39] and Gene Ontology [40]. The “Enrichment rank” means a rank in a list sorted by enrichment’s *p*-value.

Human Cell Lines Treated by Gefitinib	Interferon Alpha/Beta Signaling, Reactome ID R-HSA-909733; an Enrichment Rank/*p*-Value Adjusted	Type I Interferon Signaling Pathway, GO: 0060337; an Enrichment Rank/*p*-Value
A-431 epidermoid carcinoma, 24 h treatment	1/5 × 10^4^	8/0.005
PC-9 lung adenocarcinoma, 2 w treatment	1/10^−8^	12/0.0002
MGH119 lung non-small cell carcinoma, 2 w treatment	1/0.00003	2/0.006

## Data Availability

Genomic and transcriptomic data are available from Gene Expression Omnibus [37] project accession GSE166877. Proteomic data are available from Proteomexchange repositorium [51], project accession PXD022868.

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
