# Peer review of "Multiomic Profiling Identified EGF Receptor Signaling as a Potential Inhibitor of Type I Interferon Response in Models of Oncolytic Therapy by Vesicular Stomatitis Virus"

_ijms, 2022, doi:10.3390/ijms23095244_

Round 1

Reviewer 1 Report

The revised version of the manuscript lacks a a comprehensive bioinformatic analysis of the o-mics data. The ProteomeXchange dataset PXD022868 is not publicly available, though no password was provided to the reviewers for further review analysis. The data set is very interesting but unfortunately is not available for revision.

Author Response

We thank the Reviewer for the consideration of our work. Below we provide the responses to the specific points of the review.

Point 1. The revised version of the manuscript lacks a a comprehensive bioinformatic analysis of the o-mics data.

We thank the Reviewer for consideration, although, more specificity in the comment would be appreciated. In the work were obtained and processed two types of omics data: transcriptomic and proteomic ones. The results of original multiomics experiments led us to the hypothesis, which was further validated experimentally on the cell models in the follow-up part of the work described in Sections 2.6-2.10. Of course, while the data were analyzed comprehensively, not all results received from transcriptomic and proteomic analysis were relevant to the cell models used in the follow-up efforts. For example, an activation of immune cytokine signaling can activate targets in vivo, but not in the cell cultures. We explicitly stated this in sections 2.5 of the revised manuscript. Analysis of both RNA editing and proteomic data performed initially did not add not much to understanding the mechanisms of differential type I interferon response to the virus of interest. However, all the results of these data processing were provided. Further, analysis of the correlation between transcriptomic and proteomic data helped to determine differentially expressed transcripts, as described in Section 2.4. Note also that while most of the conclusions were drawn from the results of transcriptome analyses, we report also the results on RNA editing and proteomic data processing in spite they did not add much to the picture. We would like to have the Reviewer’s advice on how to proceed further with the results of the bioinformatics analyses in addition to the ones already reported.  

Point 2. The ProteomeXchange dataset PXD022868 is not publicly available, though no password was provided to the reviewers for further review analysis. The data set is very interesting but unfortunately is not available for revision.

It was a misprint from our side. The needed information is provided below. Please note that proteomic data in our study were merely supplementing to the transcriptomic data. The latter were mostly instrumental for understanding the mechanisms of the differential type I interferon response to the vesicular stomatitis virus. 

http://proteomecentral.proteomexchange.org/cgi/GetDataset?ID=PXD022868

Project accession: PXD022868

Project DOI: Not applicable

Reviewer account details:

Username: reviewer_pxd022868@ebi.ac.uk

Password: QW679OBz

Reviewer 2 Report

Oncolytic virus therapy is an important therapeutic tool with the potential to revolutionize cancer treatment. The variability in response to such therapy is an impeding factor that needs to be addressed. In the manuscript titled “Multiomic profiling identified EGF receptor signaling as a potential inhibitor of type I interferon response in models of oncolytic therapy by vesicular stomatitis virus” the authors explore the mechanisms that can provide a mechanistic explanation for the differential sensitivity to oncolytic virus VSV. Two cell lines displaying variable susceptibility to VSV upon interferon treatment were utilized for this study. The authors employ transcriptomics and proteomics to identify determinants of sensitivity to VSV upon interferon treatment. The authors claim increased protein levels of EGFR and HER2 mediate sensitivity to VSV upon interferon treatment. If true, the expression levels of HER2 and EGF signaling can be used as potential biomarkers for oncolytic therapy.

            Overall the study is well executed and described with ample detail in the manuscript and will be suitable for the readers of IJMS. The study however falls short of providing convincing evidence for the role of EGFR and HER2 in mediating sensitivity to VSV upon interferon treatment. The central claim made by the authors is that higher expression levels of HER2 correlate with sensitivity to VSV in presence of interferon treatment. One key experiment is missing from the manuscript. If the central hypothesis is true, overexpression of HER2 in DBTRG-05MG cells (and in other cell lines that exhibit resistance to VSV after interferon treatment) should increase their sensitivity to VSV after interferon treatment. The inclusion of this experiment will make the paper more convincing.

Author Response

We thank the Reviewer for the consideration and positive opinion of our work. Below we respond to the specific point of the review.

Point 1. One key experiment is missing from the manuscript. If the central hypothesis is true, overexpression of HER2 in DBTRG-05MG cells (and in other cell lines that exhibit resistance to VSV after interferon treatment) should increase their sensitivity to VSV after interferon treatment. The inclusion of this experiment will make the paper more convincing.

We thank the Reviewer for the thoughtful consideration. We largely agree with the argument that an artificial increase in HER2 expression would provide more direct evidence for the contribution of EGFR signaling activation to oncolytic viral therapy responsiveness. Accordingly, during our study, we attempted to conduct a similar experiment by overexpressing HER2 in glioblastoma cells that retained an antiviral response to IFN treatment. Unfortunately, the central model cell line of our study, DBTRG-05MG, was highly resistant to the genetic transduction required for the HER2 overexpression. The same was true for cultures of normal astrocytes. Please find a detailed description of transfection experiments in Supplementary File 1 added. We also briefly describe the situation with transfecting DBTRG-05MG in Section 2.10, lines 425-436 of the revised manuscript.

In addition, we would argue that the results of experiments with overexpression do not allow for their reliable interpretation since the biological effect largely depends on how much the level of expression is physiologically acceptable.

We have now begun a separate and a more comprehensive study to elucidate how activation levels of the EGFR signaling pathway affect the functioning of interferon-dependent mechanisms in several cell types. These studies include not only HER2 overexpression but also downregulation by RNA interference and gene knockouts.

In the meantime, we think that we revealed an important relationship between HER2 activity and the sensitivity of tumor cells to the oncolytic virus in the present study. Furthermore, we believe that the protection of tumor cells from the oncolytic virus that we have identified as a result of treatment with gefitinib may be of great clinical importance since it indicates the incompatibility of oncolytic viral therapy with the use of HER2 inhibitors.

We have added a few words on this subject to the concluding remarks, lines 629-636.

We sincerely think that the study's results and conclusion would help others consider more studies on the combinatorial use of current therapeutic schemes with oncolytic virus therapy.

Round 2

Reviewer 2 Report

Although, a technical difficulty precluded direct test of the hypothesis, the authors have addressed this caveat in the revised manuscript. I now recommend acceptance of the manuscript.

This manuscript is a resubmission of an earlier submission. The following is a list of the peer review reports and author responses from that submission.

Round 1

Reviewer 1 Report

Major comments

In the manuscript the authors described the analysis by MS based proteomics of whole cell extracts of two cell lines in response to type I interferon. In Tables 3 and 4, the authors have analysed the MS data from each cell line (control vs treatment) individually, once they have observed that the differentially expressed genes were not in a comparable scale for both cell lines. To compare the MS data from the two cell lines the authors can perform quantile normalization using for example R packages.

Based on the multiomics analysis, the authors validated by RNA-Seq EGF receptor genes of interest and genes involved in type I interferon, without observed differences between control and treated cells. On the other hand, the authors performed exome sequencing on the cell lines. The oncolytic resistance cell line in the presence of interferon has a mutation on BRAF affecting downstream kinase dependent reactions such as MEK/ERK. Phosphorylation status can be performed by immunoassays on the downstream targets of EGFR as well as BRAF to assess the effect of the mutation of the resistance to VSV.      

Minor comments

Please indicate the MOI used in Figure 1.

Figure 9 misses statistical analysis.

In Supplementary Figure 1 the western blots miss the molecular weight of the protein standards annotation.

Author Response

Major comments

In the manuscript the authors described the analysis by MS based proteomics of whole cell extracts of two cell lines in response to type I interferon. In Tables 3 and 4, the authors have analysed the MS data from each cell line (control vs treatment) individually, once they have observed that the differentially expressed genes were not in a comparable scale for both cell lines. To compare the MS data from the two cell lines the authors can perform quantile normalization using for example R packages.

REPLY. In this study, we did both transcriptomic and proteomic profiling for our type I interferon treatment models.  Although proteomics is, expectedly, less sensitive in this context, it highlighted the enriched pathways after the treatment similar to the one found by transcriptome analysis. Importantly, we did not analyze a secretome, which can provide unconvincing results as being technically complex and problematic in case of the model under study. Please note that many differentially expressed proteins observed for the DBTRG-05MG cell line after treatment, which indicate stronger IFN response in this line, are the secreted proteins, such as lymphokines (as shown in Table 2 in revised manuscript). These proteins are not assessed by proteome analysis of the cells as they are typically present inside them at low concentrations and secreted to the media. Proteomics data, however, were used to define the cutoff for more informative transcriptomics methods (Section 2.4). For Tables 3 and 4 (both seem to be Supplementary Tables), the preliminary analysis included the quantile normalization for proteomes. However, it was not reported, as we had chosen to focus on transcriptomes for the following analysis.

Based on the multiomics analysis, the authors validated by RNA-Seq EGF receptor genes of interest and genes involved in type I interferon, without observed differences between control and treated cells.

REPLY. Yes, indeed, the Reviewer is correct. We did not observe the difference between control and treated cells, and this was exactly the hypothesis tested in the study. We suggest that the pretreatment levels of EGF receptors determine if the absence of such a difference is observed; further supported by the results of experiments  described in Sections 2.8 and 2.9.

On the other hand, the authors performed exome sequencing on the cell lines. The oncolytic resistance cell line in the presence of interferon has a mutation on BRAF affecting downstream kinase dependent reactions such as MEK/ERK. Phosphorylation status can be performed by immunoassays on the downstream targets of EGFR as well as BRAF to assess the effect of the mutation of the resistance to VSV.

REPLY. Indeed, BRAF activating mutation in the responding cell line is the first thing to have in mind for consideration because the presence of V600E BRAF must hyperactivate the MEK/ERK pathway as pointed correctly by the Reviewer. However, at second thought, the connection between this activation and the observed effect of type I IFN protection seems dubious. On the contrary, activating the BRAF/MEK/ERK axis inhibits type I IFN responses as it was described recently for melanoma model: that (citation) ”BRAF V600E -mediated MAPK pathway activation is associated in melanoma cells with IFNAR1 downregulation” [Sabbatino et al. Journal of the National Cancer Institute, 2016]. Further, in the study of dendritic cells, it was observed that (citation) “the MEK1/2-ERK pathway inhibits type I IFN production” [Janovec et al. Front Immunol, 2018]. Thus, type I IFN responses are increased in BRAF V600E mutant DBTRG-05MG cell line not due to this mutation and despite activation of the corresponding pathway. To clarify the issue and further address the Reviewer’s concern, we added the related discussion and the above references to Section 2.2 in the revised manuscript.

Minor comments

Please indicate the MOI used in Figure 1.

REPLY.  The MOI range was indicated in revised Figure 1.

Figure 9 misses statistical analysis.

REPLY. The statistical analysis was added in Figure 9 as suggested.

In Supplementary Figure 1 the western blots miss the molecular weight of the protein standards annotation.

REPLY. The approximate molecular weights were added in Suppl. Figure 1 as suggested.

Reviewer 2 Report

The authors represented a work on investigating the underlying mechanism of the differential effects towards VSV upon type 1 interferon treatment on 2 cell lines. Through various omics analyses, the authors predicted ADAR enzyme, EGFR and HER2 proteins to be involved in the said observation. There are major flaws in the current work:

  1. Lack of clinical validation and significance. The authors worked only using established cell lines but failed to show any clinical relevance of the work done. While different cell lines have different genetic backgrounds, accounting for the differences amongst them, without clinical correlations, the data is less impactful.
  2. The statistical analysis was not convincing. For example, Figure 3, despite huge error bars, the p value was stated to be significant. There are bar graphs without any p-value.
  3. The figures given were less self-explanatory and could further be improved to increase the readability of the data.

Author Response

The authors represented a work on investigating the underlying mechanism of the differential effects towards VSV upon type 1 interferon treatment on 2 cell lines. Through various omics analyses, the authors predicted ADAR enzyme, EGFR and HER2 proteins to be involved in the said observation. There are major flaws in the current work:

  1. Lack of clinical validation and significance. The authors worked only using established cell lines but failed to show any clinical relevance of the work done. While different cell lines have different genetic backgrounds, accounting for the differences amongst them, without clinical correlations, the data is less impactful.

REPLY. We agree with the Reviewer that there are no clinical samples in this study. Note that our study was hypothesis driven as the first stage of the biomarker discovery workflow. These kinds of studies are typically intended to work with clinical specimens to demonstrate the clinical significance. The latter require different experimental designs and statistically significant amounts of clinical samples.. This study is related more to cancer cell biology rather than clinical validation. Here we discovered a differential response to IFN treatment in cell line models relevant to oncolytic viral therapy. The study aimed to disclose a cellular mechanism behind the observed effect. We also agree with the Reviewer that the mechanism must be further tested in vivo, particularly in murine models and clinical samples. We started collecting patient samples to move the study into the preclinical stage to further validate our findings.  As we mentioned above, qualifying the findings for clinical significance will require statistics amounting to 100s samples from patients, which is way upstream of the studies exploring the cell lines. In summary, we are quite surprised with the comment as there is a clear difference between the hypothesis driven experiments on cell models to discover the biological effects and the large scale studies on validation of the findings at the discovery stage utilizing large amounts of clinically proven samples from patients.   

  1. The statistical analysis was not convincing. For example, Figure 3, despite huge error bars, the p value was stated to be significant. There are bar graphs without any p-value.

REPLY. We are further surprised with the comment. The concept of p-value has been criticized recently,, e.g., by Hasley L.G., Biology Letters, 2019. Data visualization itself is considered a legitimate method to convince (or not to convince) the reader. In the study, we used five different technical approaches with different statistical analyses. Notably, for the RT-PCR data depicted in Fig. 3, we employed a conventional approach that is widely used in the field. For better clarity, we added more explanations to Section 3.6 with the additional reference. Further, in Fig.3, there are bars without p-value, as the latter cannot be calculated for the ADARB2 gene product, which was not detected in one of the cell lines of interest. In case of the absence of any measurements, the p-value cannot be calculated for the obvious reasons. Relatively low RT-PCR signals explain the "huge" error bars for inducible p150 ADAR splice isoform in controls. Such low signals provide significant observational errors, yet, a depiction of data for this isoform was needed to demonstrate its induction after the IFN treatment.

  1. The figures given were less self-explanatory and could further be improved to increase the readability of the data.

REPLY. We present the graphs and diagrams which more specifically relate to the methods used. In an attempt to better explain the conclusions drawn in the paper, we introduced Fig. 7, which depicts a scheme for pathways studied in this work. We will be grateful for any specific advice on further improving this and other figures presented in the manuscript.

Reviewer 3 Report

The authors present multi mocs approach to a very important question namely the sensitivity to oncolytic viral therapies of different cell lines after interferon therapy.

The authors first present the sensitivity of the two sell lines to VSV treatment. Ten an exome analysis was preformed on the cell lines that identified several relevant mutations. The results were briefly presented but not dissected into their depths. 

Next the RNA editing footprint was investigated.  A rather unconventional approach as the ADAR enzymes could have been evaluated directly both on RNA and protein levels.

Next proteomics and transcriptomics was used. At this point the authors present an interesting approach how to correlate transcriptomics and proteomics datasets. 

After this step pathways are analysed and correlated. More and more gene names are presented and some of these even highlighted in tables and figures such as STING, IRF3, EGFR, ERBB2. The expression of these is correlating. 

Next they show that Gefitinib is enchancing the type I interferon response in VSV sensitive cell lines and is enchancing their protection to the virus. Gefitinib is an EGFR inhibitor. 

Next they show that HER2 levels are higher in cell lines that have atenuated type one interferon response to VSV infection. 

The reviewer is active in multiomics and understands that authors would like to present all their data generated during their journey tovards finding a relevant explanation to the biological problem they are investigating.

On the other hand I would dicourage the publication off all these data in the present form as the readability of the mauscript is decreasing with each additional omics experiment added.  

My sugestion to the authors is to submit majority of their multiomics experiments ad a DATA article. There are several journals that accept Data papers. Theiy are citable and independent publications.

An independent and interesting part could be the correlation issue they have investigated and seems they have solved for proteomics and transcriptomics correlation. This could be a manuscript in itself or part of the data article.

In the mean time the streamlined results about EGFR pathway and VSV/type 1 interferon interaction could stand as an independent manuscript that is refering to the multi omics datasets generated and submited to various repositories. 

I would recommend also a killer expreiment to improve the scientific value of the results. 

You can introduce with lentiviral particles the components of the EGFR signaling pathway and check if you can change the interferon1/VSV responce in the cell lines that lhave lower expression of these components. You can order ready made viral particles. The experiment could be quite straightforward.

Good luck!

Author Response

 The authors present multi omics approach to a very important question namely the sensitivity to oncolytic viral therapies of different cell lines after interferon therapy.

The authors first present the sensitivity of the two sell lines to VSV treatment. Ten an exome analysis was preformed on the cell lines that identified several relevant mutations. The results were briefly presented but not dissected into their depths. 

REPLY. Yes, indeed, the exomes were briefly described as they were already sequenced before and deposited in the Cellosaurus database. We used these data to confirm the identity of the cell lines used for the study because to avoid identity mistakes common in the field. We further agree with the Reviewer that,, e.g., the BRAF V100E mutation in the DBTRG-05MG line was not properly discussed in the context of interferon responses. We added this discussion on page 3, Section 2.2 in the revised manuscript.  

Next the RNA editing footprint was investigated.  A rather unconventional approach as the ADAR enzymes could have been evaluated directly both on RNA and protein levels.

REPLY.  We agree with the Reviewer’s note and we actually estimated the expression levels of ADAR, ADARB1, and related genes (Fig.3) before the footprint analysis of Alu repeats. However, we believe that the addition of the Alu index is needed as it provides the only way to estimate the activity of these enzymes. The Alu index method was introduced recently and quickly became a recognized method of choice in the field [Roth et al, Nature Methods, 2019].

The reviewer is active in multiomics and understands that authors would like to present all their data generated during their journey towards finding a relevant explanation to the biological problem they are investigating.

On the other hand I would dicourage the publication off all these data in the present form as the readability of the mauscript is decreasing with each additional omics experiment added.

My sugestion to the authors is to submit majority of their multiomics experiments as a DATA article. There are several journals that accept Data papers. They are citable and independent publications. An independent and interesting part could be the correlation issue they have investigated and seems they have solved for proteomics and transcriptomics correlation. This could be a manuscript in itself or part of the data article.

In the mean time the streamlined results about EGFR pathway and VSV/type 1 interferon interaction could stand as an independent manuscript that is referring to the multi omics datasets generated and submitted to various repositories.

REPLY. We fully respect the Reviewer’s opinion but cannot entirely agree with it. In the study we present quite a straightforward story which may be briefly described as follows. Omics analyses gave enrichments in the VSV resistant cell line after interferon for a group of secreted cytokines and lymphokines, importantly, with a partial reproduction of the earlier result by Grunfogel et al. (2016) (Section 2.5). This result may be further tested in vivo, as the cytokines do not act in the cell models. Instead, transcriptomics had shown a deficiency in the EGF signaling in the VSV resistant cell line, which leads to a hypothesis of a connection between the deficiency of EGF receptor, in particular, HER2, and the cell resistance to VSV (Section 2.6). The latter was supported by our results on three cell lines and also from some publicly available transcriptomics data.  Separating data by sending the omics to the other paper breaks the logic behind the origin of the above hypothesis. Thus, while we share the Reviewer’s view that a paper should illustrate and focus on a single thought, the current trend towards the multidisciplinarity of research advocates for the opposite these days. Further elaborating on this issue we note that the exile of omics results to the “data” journals would support the harmful idea of classical molecular biologists that omics is an expensive production of experimental noise. Moreover, all omics data obtained for this study will be 100% transparent, available publicly in the corresponding repositories,  and being annotated there soon in the standardized manner [Dai et al., Nat. Commun. 2021], which is functionally equivalent to the ultimate goal of the ”data” journals. To summarize, we would like to keep the integrative nature of the multi-omics data obtained for this study and leave this paper as a whole narrative.

I would recommend also a killer expreiment to improve the scientific value of the results. 

You can introduce with lentiviral particles the components of the EGFR signaling pathway and check if you can change the interferon1/VSV responce in the cell lines that have lower expression of these components. You can order ready made viral particles. The experiment could be quite straightforward.

REPLY. We thank the reviewer for this idea. Indeed, It would be an exciting and important follow-up study. We also plan to reproduce our EGF signaling-related findings with additional amounts of established and primary glioma cell lines. However, it takes time, additional funding, and significant efforts which are the focus of our follow-up project. Nevertheless, we would like to release the current results to make it possible for other researchers to use the hypotheses stated here in the complementary projects.

Round 2

Reviewer 1 Report

Even though the authors had preformed a multi-omics approach, which allows to profile the whole proteome and transcriptome,  data analysis and data integration must be further improved. Data presentation for example using heatmaps of both transcriptomic and proteomic analysis of both cell lines with and without interferon response is advisable for a clear visualization of the data.

Graphical analysis display in Figure 5 is not justified concordantly with the text description.

Reviewer 2 Report

The reviewer is in an opinion, while the authors claim the study is a “hypothesis-driven experiment on cell models”, validation even on smaller number of clinical samples and in vivo work are essential for the consideration of good journals such as IJMS. Since the authors claim that they have begun collecting clinical samples, upon validation the authors are highly recommended for the consideration for resubmission.

Reviewer 3 Report

Thank you for your answers. I cannot accept your answers.